# Single Nucleotide Polymorphism Induces Divergent Dynamic Patterns in CYP3A5: A Microsecond Scale Biomolecular Simulation of Variants Identified in Sub-Saharan African Populations

**DOI:** 10.3390/ijms22157786

**Published:** 2021-07-21

**Authors:** Houcemeddine Othman, Jorge E. B. da Rocha, Scott Hazelhurst

**Affiliations:** 1Sydney Brenner Institute for Molecular Bioscience, Faculty of Health Sciences, University of the Witwatersrand, Johannesburg 2001, South Africa; jdarocha1@gmail.com (J.E.B.d.R.); scott.hazelhurst@wits.ac.za (S.H.); 2Division of Human Genetics, National Health Laboratory Service, School of Pathology, Faculty of Health Sciences, University of the Witwatersrand, Johannesburg 2001, South Africa; 3School of Electrical and Information Engineering, University of the Witwatersrand, Johannesburg 2001, South Africa

**Keywords:** pharmacogenomics, molecular dynamics, CYP3A5, sub-Saharan Africa

## Abstract

Pharmacogenomics aims to reveal variants associated with drug response phenotypes. Genes whose roles involve the absorption, distribution, metabolism, and excretion of drugs, are highly polymorphic between populations. High coverage whole genome sequencing showed that a large proportion of the variants for these genes are rare in African populations. This study investigated the impact of such variants on protein structure to assess their functional importance. We used genetic data of *CYP3A5* from 458 individuals from sub-Saharan Africa to conduct a structural bioinformatics analysis. Five missense variants were modeled and microsecond scale molecular dynamics simulations were conducted for each, as well as for the CYP3A5 wildtype and the Y53C variant, which has a known deleterious impact on enzyme activity. The binding of ritonavir and artemether to CYP3A5 variant structures was also evaluated. Our results showed different conformational characteristics between all the variants. No significant structural changes were noticed. However, the genetic variability seemed to act on the plasticity of the protein. The impact on drug binding might be drug dependant. We concluded that rare variants hold relevance in determining the pharmacogenomics properties of populations. This could have a significant impact on precision medicine applications in sub-Saharan Africa.

## 1. Introduction

The fate of drug molecules is determined by a set of biological processes controlling the Absorption, Distribution, Metabolism, and Excretion (ADME) in the organism. Upon administration, drugs can induce adverse effects which include toxicity. The balance between the induction of the therapeutic effect and undesirable outcomes depends on the set of proteins encoding the ADME functions. Among these proteins, the CYP P450 superfamily is an important group of enzymes which play vital roles in Phase I metabolism of multiple drug types. Most biocatalyzation reactions are processed by the CYP3A subfamily, with an estimated fraction of 24% of the total number of chemicals [1]. In particular, *CYP3A5* is a highly polymorphic gene with 29 total variants belonging to 9 star alleles (https://www.pharmvar.org/gene/CYP3A5, accessed on 1 July 2021) [2]. *CYP3A5*3* results in a defective mRNA and is perhaps the most relevant allele for phenotype classification [3]. The *CYP3A5*3* splice variant has been studied in African populations regarding ritonavir [4] and artemether [5] drug response, but studies of *CYP3A5* missense variants in these populations remain scarce. Other variants of clinical importance include *CYP3A5*6* and *CYP3A5*7* which cause a protein truncation and a reading frame shift respectively [3]. The frequencies of these three major alleles are population dependant, particularly on ethnicity or ancestry. Allele frequencies range between 4 and 81%, 5 and 25%, and 0 and 21% respectively for *CYP3A5*3*, *CYP3A5*6*, and *CYP3A5*7* in Africa [6]. There have been some conflicting studies about the genotype-phenotype association of *CYP3A5*. A recent systematic review and meta-analysis showed that *CYP3A5* genotype does not correlate uniformly with pharmacokinetics properties of tacrolimus [7]. Variant levels of significance response have been noted between different populations as well as within populations of the same ethnic composition.

More than 280 drugs (Appendix A) are known to interact with CYP3A5 including artemether for the treatment of uncomplicated *Plasmodium falciparum* infections [8] and ritonavir as a protease inhibitor for HIV [9]. Our recent analysis of the ADME genes from the genomes of 458 individuals from sub-Saharan Africa showed a high level of diversity even between populations residing at close geographical proximity [10]. Moreover, we also found that a large proportion of rare variants might be determinant for diversifying drug-response in African individuals. From these observations, we coined the term *“genetic diversity bottleneck in precision medicine"* to express the difficulty in accounting for low levels of granularity imposed by the genetic diversity when setting up targeted healthcare strategies.

Conceptual views of the functional implication of genetic variants have been long perceived as a binary categorization to be either deleterious or neutral. In particular for pharmacogenomic studies, this simplistic view offers limited insight into molecular mechanisms that control drug processing function. So far, the significant impact of rare variants in pharmacogenomics was only theorized in the context of genomic studies. Sequencing more African individuals will enrich the repertoire of rare variants with potentially high impact. The extent of their effect at the gene product level is poorly studied and such information is necessary to establish prioritization plans of variant actionability that is not based solely on the prevalence in the population. In addition, considering a single pharmacogene, the evaluations of the generic effect of rare variants on different drugs is valuable information in defining the spectrum of action in diagnostics and treatment.

To evaluate the extent of rare variants genetic diversity affecting the gene product functionality, we extended our analysis by focusing on *CYP3A5* gene based on the data from high coverage whole genome sequencing from individuals from sub-Saharan Africa (we call this the high coverage African ADME Dataset—HAAD), combined with 1000 Genome Project data. We then used molecular modeling and molecular dynamics to evaluate the effect of single amino acid variants on the conformational properties of protein structure. We also studied the effect on the binding of artemether and ritonavir as proof of concept.

## 2. Materials and Methods

### 2.1. Samples and Genome Analysis

The genetic characterization of *CYP3A5* gene was derived from our previous study, which included the analysis of high coverage whole genome sequencing data from 458 samples from 15 African countries. These data were collected by the H3Africa/GSK ADME consortium. This analysis also includes data from 507 African samples from the *1000 Genomes* initiative. Variants were identified in jointly called HAAD and *1000 Genomes* datasets. For the sake of brevity, here we only give the outlines of the genome analysis method but a detailed description is extensively exhibited by da Rocha et al. [10]. The GrCh37 assembly was used as a reference for the genome coordinates and the gene transcripts. The variants of *CYP3A5* gene were annotated using the ‘g_miner’ workflow (https://github.com/hothman/PGx-Tools/tree/master/workflows/g_miner, accessed on 1 July 2021). Variant Effect Predictor (VEP) [11] was used for the functional annotation of the missense variants that combines different annotation and prediction algorithms. We have also integrated the predictive approach by Zhou et al. [12] developed specifically for the annotation of pharmacogenes. Allele frequencies of *CYP3A5* were calculated using PLINK version 1.9 [13].

### 2.2. System Preparation

We have simulated three different systems of CYP3A5 protein. These include: a substrate free enzyme (PDB code: 6MJM) [14], the complex with ritonavir (PDB code 5VEU) [15], and the CYP3A5/artemether complex generated by docking. The latter also uses the PDB entry 5VEU for the structure of the enzyme. Missing loops (segments 260–270, 280–288 in the substrate free structure and segment 282–288 in the bound structure) were generated with the loop modeling routine from MODELLER software version 9.22 [16]. Five hundred conformations were generated and the structure with the best DOPE score [17] was retained for the further steps. We estimated the ionization states at pH of 7.4 of all the charged groups of CYP3A5 using PROPKA3 [18] according to which we neutralized residues Asp^182^, Asp^269^, and Glu^294^.

### 2.3. Force Field Parameters for Heme Group, Ritonavir, and Artemether

Parameters for the CYP3A5 amino acid residues are derived from the AMBER ff14SB force field which is in better agreement with experimental results to simulate cytochrome P450 proteins [19]. General Amber Force Field (GAFF) [20] atom types were assigned to the heme group and parameters from Shahrokh et al. [21] were used to account for the bonded configuration between the Cys^441^ sulphur atom and the Fe metal ion as well as the particular electronic state of the heme group. For the substrate-free enzyme, we have used the parameters of compound I characterized by the formation of a covalent bond between an oxygen atom and the heme iron as it was seen in the crystal structure [22]. For the bound configuration with ritonavir and artemether, we used the parameters of the pentacoordinate ferric high-spin configuration of the heme group. Parameters from the GAFF force field were assigned to both ritanavir and artemether ligands and the partial charges were calculated by the AM1-BCC method implemented in the antechamber tool [23].

### 2.4. Artemether Docking to CYP3A5

The molecular docking approach is used to predict the bound form between a ligand and a target (commonly known as a receptor) when a reliable experimental structure of the native complex is not available. Docking algorithms sample the translational and rotational degrees of freedom of a ligand over a 3D domain space of the receptor based on an energy scoring function. Autodock Vina [24] was used to predict the complex between artemether and CYP3A5. The partial charges calculated by antechamber were assigned to the structure of the ligand in ‘PDBQT’ format. Partial charges calculated by Shahrokh et al., were assigned to the heme group atoms et al. [21] as part of the receptor structure. This method has been shown by the authors to significantly improve the performance of the docking. Gasteiger charges were assigned to all amino acids of CYP3A5. The docking was conducted in a grid box that covers the entire catalytic pocket of the receptor. We executed five runs of docking starting from different seeds to ensure the consistency of the results. The degree of exhaustiveness was set to 200 while keeping active all the rotatable bonds of artemether. From the nine binding modes that were returned, we selected those that interact directly with the heme atoms, after which we ran 100 ns of molecular dynamics simulation. The choice of a reasonable binding model is based on five properties that include the fraction of native contacts, the distance of the center of mass to the iron atom of the heme group, and the Root Mean Square Deviation of all atoms in the ligand compared to the docking pose.

### 2.5. Generating the Structures for the Variants

We have generated five structures incorporating the variants from the combined H3Africa and 1000 Genomes project data as well as for the Y53C variant (corresponding to the *CYP3A5*11* allele) for which there is strong biochemical evidence about its role in reducing the catalytic capacity of the enzyme [25]. Structures of the variants were generated by removing all uncommon atoms that have different atom names of the side chains with the reference position. The amino acid’s name is then changed to its corresponding variant. We then used the software ‘leap’ from AMBER 18 [26] to build the missing amino acid atoms. Initial refining included removal of severe steric clashes by running two in vacuo energy minimization cycles corresponding to 250 steps of the steepest descent stage followed by 400 steps of conjugate gradient minimization. During the run, a restraint force constant of 2 kcal/mol/Å^2^ was applied on all the atoms of the system except those belonging to the variant position. The same routine was also applied to the reference structure (wild type form).

### 2.6. Molecular Dynamics

Molecular dynamics experiments involve the in silico simulation of a protein structure over a time based trajectory. The molecular dynamics simulation aims to sample conformations of highly populated states of a protein. The sampling constructs a trajectory of conformations that are representative of an energy landscape. The latter is altered between the CYP3A5 variants, subject to this analysis. Interpretations of variant impact can be made by assessing structural changes throughout the trajectory. These can be used to better our understanding of variant impact on structural stability and ligand interaction, which may have implications for drug response. Prior to running the simulation, the system was first solvated with a TIP3P octahedron box. Chloride counter-ions were added conveniently in order to neutralize the charge of the system. The distance between the edges of the simulation box and the solute atoms was set to a minimum of 15 Å. Before the execution of the molecular dynamics simulation, two stages of energy minimization were run to remove the steric clashes resulting from adding the solvent molecules to the system. In the first stage, we used a steepest descent integrator for 500 steps followed by 9500 conjugate gradient minimization while restraining all the atoms of the solute with a force of 500 kcal/mol/Å^2^. In the second stage, we run two consecutive steepest descent and conjugate gradient minimizations for 1000 and 6000 steps respectively. The refinement was run under implicit solvent conditions by activating the Hawkins, Cramer, Truhlar Generalized Born model.

The molecular dynamics simulation begins with heating the system from 50 K to the desired temperature of 300 K. We have fixed the random seed number (ig) to 96,465. Restraining forces were applied on all the atoms of the solute at a value of 10 kcal/mol/Å^2^. A Langevin thermostat with a collision frequency of 5 ps^−1^ was applied to control the temperature fluctuation around 300 K. The heating stage was run for a total time of 20 ps after which we run an Isothermal–isobaric (NPT) simulation. At this stage, we used a restraining force of 10 kcal/mol/Å^2^ applied on all atoms of the solute. Then, the force was gradually lifted by a decrement of 2 kcal/mol/Å^2^ until the system was equilibrated. Each of the stages is run for a total time 80 ps. The final production phase was operated in the canonical ensemble (NVT). We generated a continuous trajectory spanning a simulation time of 1.5
μs for the unbound CYP3A5 structure. During all the simulation stages, periodic boundary conditions were applied. The Particle Mesh Ewald approximation was used for efficient calculation of the electrostatic interactions. We also used a cutoff value of 12 Å for the nonbonded interactions. The SHAKE algorithm was applied to allow for an integration time of 2 ps and snapshots are collected at intervals of 10 ps.

We have used a slightly different protocol for the molecular dynamics simulation of the CYP3A5/artemether and CYP3A5/ritonavir complexes by running three independent simulations each with a total production time of 200 ns. The random seed number for the three trajectories were set to 18,917, 74,269, and 96,465.

### 2.7. Analysis of Molecular Dynamics Simulation

An in house script was employed to analyze the data from the molecular trajectory simulations. The code is available at https://zenodo.org/record/4548257/files/othman_md_lib.py, accessed on 1 July 2021. The library is written in python and uses mdtraj [27] and pytraj [28] modules for the development of routines and functions.

#### 2.7.1. Essential Dynamics Analysis

Extracting relevant information from an ensemble of molecular dynamics conformations is facilitated by essential dynamics methods. These aim to reduce the dimensionality of the dataset and to detect biologically relevant patterns of collective motions. In doing so, we applied principal component analysis for the conformations at the equilibrated phase of the trajectory. The ensemble of snapshots was first fitted to the crystal structure of CYP3A5 to remove the degrees of freedom related to transitions and rotations. The low dimension components were calculated by the ‘pca’ method from ‘pytraj’ to return the corresponding eigenvalues and eigenvectors as well as the projection of the atomic coordinates into the lower-dimensional subspaces. Only $C_{\alpha }$ atoms were considered for the calculations of the Principal Components (PCs).

The Root Mean Square Inner Product (RMSIP) [29] was calculated to evaluate the overlap between the wild type and the variant conformational ensembles according to the following formula:$$RMSIP=\sqrt{\frac{1}{20}\sum_{i=1}^{20}\sum_{j=1}^{20}{\left(\mu_{i}\;\cdot \;\nu_{i}\right)}^{2}}$$

Here, $\mu_{i}$ and $\nu_{i}$ correspond, respectively, to the first 20 eigenvectors calculated from the wild type and the variant trajectories.

Elements of the dynamic cross-correlation matrix were calculated according to the following formula.
$$DCC\left(i,j\right)=\frac{\langle \Delta \;r_{i}\left(t\right)\;\cdot \;\Delta r_{j}\left(t\right)\rangle }{\sqrt{\langle \Delta \;r_{i}\left(t\right)\;\cdot \;\Delta \;r_{i}\left(t\right)\rangle \langle \Delta \;r_{i}\left(t\right)\;\cdot \;\Delta \;r_{j}\left(t\right)\rangle }}.$$$\Delta \;r_{i}\left(t\right)$ and $\Delta \;r_{j}\left(t\right)$ are displacement vectors at time *t* calculated by subtracting the coordinates vector of an atom *i* or *j* from the average coordinates calculated over the entire ensemble.

#### 2.7.2. Free Energy Landscape

The Free Energy Landscape (FEL) can be established using the density data of reaction coordinates to explore the energy property of a protein in a three-dimensional space. FEL was obtained according to the following equation:$$\Delta \;G=-k_{b}T\ln P\left(X\right)$$
where $k_{b}$ corresponds to the Boltzmann constant, *T* is the absolute temperature and *P*(*X*) is the probability of the reaction coordinate *X* obtained by binning the distribution of the data. In our study we have used the projections of $C_{\alpha }$ atom coordinates over PC1 and PC2 as reaction coordinates.

#### 2.7.3. Binding Free Energy Calculation

The binding energy between a ligand and a receptor was estimated according to the formalism established by Duan et al. [30]. The method deviates slightly from the original scheme [31] by stating that the binding energy is the outcome of a linear combination between a gas term ($\Delta \;G_{\mathrm{gas}}$) and a solvation term ($\Delta \;G_{\mathrm{solv}}$).
$$G_{\mathrm{binding}}=\Delta \;G_{\mathrm{gas}}+\Delta \;G_{\mathrm{solv}}$$

The entropic contribution makes part of the gas term according to the following equation.
$$\Delta \;G_{\mathrm{gas}}=\langle E_{\mathrm{pl}}^{\mathrm{int}}\rangle -T\Delta \;S$$$\langle E_{\mathrm{pl}}^{\mathrm{int}}\rangle$ is calculated by averaging the interaction energy between the ligand and the protein over the conformations constituting the ensemble of molecular dynamics trajectory, here:$$-T\Delta \;S=k_{b}T\;\ln\left[\langle e^{\beta \;\Delta \;E_{\mathrm{pl}}^{\mathrm{int}}}\rangle \right]$$$k_{b}$ is the Boltzmann constant, *T* is the absolute Temperature and $\beta \;=1/(k_{b}T)$. $\Delta \;G_{\mathrm{gas}}$, $\Delta \;G_{\mathrm{sol}}$ and $\langle E_{\mathrm{pl}}^{\mathrm{int}}\rangle$ can be calculated by the MMPBSA.py program [32] from AMBER molecular dynamics package.

#### 2.7.4. Cavity Volume and Tunnels Geometry Calculations

To compute the volume of the catalytic binding site of CYP3A5 we employed the program ‘mdpocket’ [33]. Additionally, we used caver 3.0.3 [34] to calculate the geometry of the tunnels within the enzyme used by the drugs to enter to the catalytic binding site. The starting point coordinates were set to the position of the iron atom in the heme group. The probe radius, the shell radius, and the shell depth were set to values of 0.9, 5, and 4 respectively and we used a threshold of 3 for the hierarchical clustering of the tunnel’s geometries.

## 3. Results

### 3.1. Genetic Characterization of CYP3A5 Variants from WGS

The genome analysis and variant annotation of the combined High Coverage African ADME dataset (HAAD) and 1000 Genomes Project (KGP) datasets identified 524 variants of which a large proportion corresponds to intronic and noncoding regions. A frameshift variant (rs41303343-A), caused by a deletion, has a frequency of 0.113 in African populations. The variant is identified in *CYP3A5*7* and is associated with defective metabolism of many drugs including tacrolimus and ritonavir. Only five missense variants were identified in the source African populations of this study; all are rare with frequencies below 0.006 (Table 1). None of these variants have been reported by ClinVar.

These variants were mapped to the protein structure of CYP3A5 (Appendix A). Except for amino acid substitutions R28C and V238A, all the other variants are located in the buried core of the protein. V238A belongs to the F/G loop of CYP3A5 that controls the access to the catalytic binding site [35].

### 3.2. Molecular Dynamics Simulation of CYP3A5 Variants

Molecular dynamics simulation was run for the wild type and the variant forms of CYP3A5 in the absence of the substrate (Unbound structure) for a total production time of 1.5
μs.

#### Overall Effect of the Variants on CYP3A5 Structure

The convergence of molecular dynamics trajectories was achieved (Appendix A). We calculated the Root Mean Square Deviation (RMSD) along the entire production time. We excluded the loop segments that were missing in the crystal structure of CYP3A5 from the RMSD calculation because their high flexibility may prevent the detection of any relevant structural divergence located on other parts of the protein. The RMSD calculation considered the unbound crystal structure of CYP3A5 (PDB code: 6MJM) as a reference. The trajectory of the wild type structure was equilibrated at around 200 ns and shows a stable behavior along the rest of the simulation time with short plateau phases that indicate the sampling of different local energy minima (Figure 1). Unless mentioned explicitly, all further analysis uses the snapshots from the equilibrated phase of the molecular dynamics trajectory. All the variant structures show little deviations from the WT RMSD profile and values remain below 0.2 nm. In particular, the Y53C variant which was found to decrease the activity of CYP3A5 [25] has no significant deviation compared to the WT backbone atoms. The largest deviation was observed for the V238A variant averaging 0.5 Å along the full simulation time.

To assess the local fluctuation of CYP3A5 segments, we calculated the Root Mean Square Fluctuation (RMSF) for the $C_{\alpha }$ atoms (Figure 2). The highest fluctuation values correspond to the loop regions 260–270, 280–288 which is expected because the electron density map in the protein structure was not solved for these segments. All other parts of CYP3A5 do not show a major deviation from the wild type structure except for the F/G loop segment spanning the 210–240 amino acids. The divergent fluctuation pattern for these residues differs between the variants. L82R seems to be the least affected given that the local differences in fluctuations are low compared to the other variants. The extent of the divergent fluctuation at the F/G loop is limited to segment 210–230 for Y53C, I149T, I276T variants while it is much wider for R28C and V238A.

### 3.3. Analysis of the Collective Motions within CYP3A5

Dynamic Cross-Correlation Matrices detect blocks of residues that move collectively in the 3D space. A collective motion can be either correlated or anticorrelated. We focused our analysis on pairs of residues that show correlation values of more than 0.5 or less than −0.5. We noticed that variants R28C, Y53C, L82R, and V238C display differences in the DCCM compared to the wild type form (Figure 3) characterized by an increase in the intensity of some zones in the DCCM and a dense network of connectivities between the $C_{\alpha }$ atoms. The analysis revealed four regions of CYP3A5 that are most affected by the single amino acid substitution: The F/G loop, the F and G helices, the N-terminus domain, and the I and G helices. The R28C variant shows a dense network at the N-terminus domain. The motion of the region 291–302 of the I-helix is correlated with the N-terminus part of the G helix spanning residues 243–252. The DCCM analysis shows also correlated dynamics between the entirety of the F helix and the first three N-terminus turns of the G helix. The same type of collective dynamic was also observed in L82R; however, it was not as important as for Y53C. V238A variant shows an increase in the network density for the F/G loop and between the F and G loops. Moreover, with five residues belonging to the B/C loop, the correlation network established with the F/G loop is extensive compared to the wild type form which involves only three residues.

### 3.4. Essential Dynamics of CYP3A5

The analysis, up to this point, has shown local differences between the different variants and the wild type forms. We then asked the question of whether the single variation in the amino acid sequence of CYP3A5 can lead to a significant effect on the essential conformational modes. The $C_{\alpha }$ atom coordinates of each snapshot of the trajectories were projected onto PC1, PC2, PC3, and PC4 subspaces. Because the flexible loops of CYP3A5 can be dominant in the calculated modes, we choose to discard them from the PCA analysis. We noticed that the first 7 PCs are different between the analyzed variants as shown from their divergent weights in the explained variance. The first four PCs account for 50% of the explained variance for the wild type form (Appendix A). The value increases to 63% and 61% for Y53C and V238A variants. It remains below 57% for all the other variants. Most of the differences are found for the first PC (PC1). For the wild type form, it contributes to 19% of the explained variance. I149T and R28C show close values to the wild type form. Y53C and V238A show the highest differences with values of 34% and 25% respectively for PC1.

To estimate the overlap between the macromolecular modes, we calculated the RMSIP between the first 20 eigenvectors of the wild type form and each of the six variants. The RMSIP values can be rounded to 0.70–0.8 for all the variants which shows a significant overlap between the trajectory of the wild type form and the other variants. The PC1 vs. PC2 plot shows however some disparities between the two major dynamic modes (Figure 4). The molecular dynamics snapshots of R28C, Y53C, L82R, V238A, and I276T are extended to a wider range in the 2D phase space compared to the wild type form. In general, the variance is more important for PC1. On the other hand, PC3 vs. PC4 plot does not differ widely between the wild type and the mutant (Appendix A).

The amplitude and the direction of movement of each of the $C_{\alpha }$ atoms were estimated by projecting the coordinates on the first major eigenvector that explains the variance of the dynamics. Porcupine plots (Appendix A) show that residues belonging to the segments controlling the access to the catalytic site are the most major contributors in the protein motion for all the variants. However, the amplitude and the direction of these amino acids differ significantly from one variant to another and compared to the wild type form. Both Y53C and V238A variants showed a wider divergence from the wild type form. The amplitude of displacement is important and mainly affects the F/G loop segment.

### 3.5. Analysis of the Free Energy Landscape

Next, we asked the question of whether the variation in the amino acid sequence of CYP3A5 can affect the energy property of folding which would be best assessed at the global minimum of the Free Energy Landscape (FEL). To proceed, we choose PC1 and PC2 as the reaction coordinates of the FEL to ensure better discrimination between the different protein states. We noticed that the free energy surface differs significantly in terms of topology between the different assessed variants. Compared to the wild type, Y53C, I149T, V238A, and I276T have many local energy wells that have a depth close to the global energy (Appendix A). To appreciate the differences between the conformation at the global minimum of each trajectory, we extracted them from each FEL then projected their coordinates onto the subspaces described by the first and the second eigenvectors of the wild type form. In Figure 5, we show the FEL of the wild type form in 3D representation and its corresponding 2D projection. We noticed that all the conformations at the global minimum of each variant are located away from the position of the global energy well of the wild type form. Some of the conformations are even positioned at higher energy levels. Structural fitting to the wild type conformation at the global minimum shows a large deviation mainly at the F/G loop, the F and G helices, and the B/C loop.

### 3.6. Catalytic Pocket Volume Calculation

We desired to verify the impact of the genetic diversity on the geometry of the catalytic pocket of CYP3A5 (Figure 6A). All the simulated variants show differences from the wild type form with similar values of variance (Figure 6B). The Y53C variant shows the least difference with an average volume of 2019.04 Å^3^ which is slightly larger than the volume of the wild type pocket measuring 1970.70 Å^3^. L82R and I276T show close values of 1805.27 Å^3^ and 1793.63 Å^3^ respectively. Moreover, R28C, I149T, and V238A have the smallest catalytic pocket volume with values of 1625.61 Å^3^, 1479.37 Å^3^, and 1438.22 Å^3^ respectively which represent a shrinkage of 17%, 24%, and 27% respectively relative to the volume calculated for the wild type form. The CYP3A subfamily was shown to be promiscuous allowing the accommodation of a large repertoire of drugs in the catalytic pocket of the enzymes. The wild-type pocket volume range of CYP3A5 calculated in our study confirmed such property. Our results also show higher pocket volume values reported previously for CYP3A4 evaluated to 1386 Å^3^ and 520 Å^3^ from two different studies [36]. These values, however, were reported based on single-point calculations for the unbound structure of CYP3A4. Another calculation obtained a value of pocket volume of 765 Å^3^ of CYP3A4 bound to ritonavir which is almost two times less than the lowest volume calculated to any of CYP3A5 variants in complex with the same drug [37].

### 3.7. Binding Free Energy Estimation of CYP3A5/Ritonavir and CYP3A5/Artemether Complexes

To assess the effect of CYP3A5 variants on its binding capacity to drug molecules, we studied the dynamics of the enzyme in its bound form with ritonavir and artemether having 50 and 21 heavy atoms respectively. They were selected to test the effect on the binding of small and large size ligands to the catalytic binding pocket. We have obtained a reasonable solution for the CYP3A5/artemether predicted by molecular docking. The complexes were used to run three independent molecular dynamics simulations for an accumulated time of 600 ns for each variant. The binding free energy included the snapshots collected from the last 100 ns for each independent trajectory (a total time of 300 ns). We have verified the convergence of the calculation using this approach and we have put more details in Appendix A.

According to the MM-GBSA energy (Table 2) estimated for CYP3A5/ritonavir complex, the drug binds to all the forms with favorable energies except for V238A where $\Delta \;G_{\mathrm{binding}}=1.44$ kcal/mol. Y53C and I149T also show weak binding energy values while the affinity of the drug is better for R28C, L82R, and I276T compared to the wild-type form. In general, the variance of the $\Delta \;G_{\mathrm{sol}}$ term is less important compared to the entropy. V238A showed the least favorable $\langle E_{\mathrm{pl}}^{\mathrm{int}}\rangle$ among all the other variants at −55.97 kcal mol^−1^. Y53C and V238A show the least favorable entropy terms evaluated at 21.08 kcal/mol and 19.45 kcal/mol respectively.

Figure 7A shows the variation of the estimated MM-GBSA energy as a function of the cavity volume calculated from the previous analysis. Relative to the range of the calculated binding energies, extreme values correspond to either small or large pocket volume. We have fitted these data to a polynomial regression model of the form y=(x−a)(x−b)(x−c) where a = $1.33\times 10^{-4}$, b = −0.468, c = $3.98\times 10^{2}$ for ritonavir and a = $-6.25\times 10^{-5}$, b = 0.215, c = $-1.95\times 10^{2}$ for artemether. A strong coefficient of determination ($R^{2}$) of 0.94 was noticed for ritonavir. The binding of artemether to CYP3A5 is less favorable to all variants ranging from −15.45 kcal mol^−1^ for I149T to −8.22 kcal mol^−1^ for R28C (Table 2). The variance of $\Delta \;G_{\mathrm{binding}}$ is less important compared to the values estimated for ritonavir. The variance is even lower considering $\langle E_{\mathrm{pl}}^{\mathrm{int}}\rangle$ energy term which ranges from −29.31 to −33.16 kcal mol^−1^. In addition, −TΔS and $\Delta \;G_{\mathrm{solv}}$ energy terms equally contribute to the unfavorable energy penalty in all variants. For instance, we estimated similar values of 12.08 kcal/mol and 12.85 kcal/mol respectively for −TΔS and $\Delta \;G_{\mathrm{solv}}$ energy terms for the R28C variant. Moreover, the fitting of a polynomial regression model between the catalytic pocket volume and $\Delta \;G_{\mathrm{binding}}$ for artemether shows a weaker $R^{2}$ of 0.73 compared to ritonavir (Figure 7B).

### 3.8. Analysis of the Tunnels

Analysis of CYP3A5 tunnels allows for the evaluation of the effect of the dynamics on the drug pathways that facilitate the transition of the drug to the catalytic site. Figure 8 shows first four clusters identified by Caver 3.0 according to their priority values—the higher the priority the more important is the tunnel. Indeed, all the simulated systems show a distinguished set of tunnels calculated from the microsecond level simulation. In general, the opening gates to the catalytic site are located either in the B/C loop side or between the F/G loop, the 477–483 segment, and the 49–57 segment (upper entry). The distribution of the tunnels differs considerably between the variants. For instance, while the wild type form has a single gate, G1, located near the B/C loop, variants R28C, Y53C, L82R I149T, and I276T have multiple gates. Moreover, Y53C has a single gate at the upper entry contrary to the wild type and other variants that possess more than one gate. At the quantitative level, we also noticed that there is no favorite common gate between the evaluated variants. For instance, the highest priority $P_{C1}$ is equal to 0.64 and corresponds to the G4 gate in the wild type form, while $P_{C1}$ is equal to 0.65 in I149T and corresponds to G5 gate located nearby the B/C loop.

## 4. Discussion

Understanding the pharmacogenetic landscape within a particular gene is a key step for establishing targeted clinical interventions in precision medicine. Variant interpretation, however, is still a limiting factor to acquire a better insight into the genotype–phenotype relationship and its implication in an individual’s responses to drugs. This is mainly the cause of experimental difficulties in collecting data at the molecular level of the gene product. Thus, multiscale modeling is a valuable tool to predict the molecular impact of genetic diversity at the protein level and to understand the molecular mechanisms of the phenotype expression.

None of the studied variants induces a large structural drift compared to the wild-type form even after 1.5 μs of simulation including Y53C which has been shown to reduce the activity of CYP3A5. This is unexpected since the mapping of the variants on the protein structure suggests that some of the variable amino acids might indeed have a destabilizing effect such as the case of L82R representing a substitution of a buried hydrophobic residue with a charged amino acid. Indeed, this is consistent with previous results suggesting the high level of structural conservation within the Cytochromes P450 superfamily [38]. It appeared, however, that most of the divergence between the variants is mainly affected by the B/C loop and F/G loop dynamics as shown by the local fluctuations.

We, therefore, hypothesized that the main impact of the genetic variability affects the plasticity of different structural elements that are critical for different steps of the catalysis process. These include the controlling of the entry of the drug to the channel that leads to the catalytic binding site, the binding to the catalytic pocket, and the transport process within the enzyme.

The evidence presented here shows that the genetic variability of CYP3A5 affects the structural plasticity of the protein specifically the region spanning the B/C helices, the F/G helices, and the upper roof segments relative to the heme group. Different modes of dynamics for the F/G loop might have an impact on the way each form of CYP3A5 responds to the triggering of the catalysis process. Diversification in the low-frequency dynamic modes revealed by essential dynamics analysis might be the consequence of the different observed correlated motion patterns. These outcomes result in a significant effect to shift the dynamic equilibrium of CYP3A5. This was supported by the establishment of the free energy landscape using the two first major PCs as reaction coordinates. The major conformational events revolve around the structure at the global minimum. Therefore, the different topologies of the FEL, as well as the divergent conformations of the structure at the global minima, revealed that for each of the CYP3A5 variants, different levels of energy barriers have to be crossed by the protein to fulfill its function. Considering all that, we also assume that the conformational properties, encoded by the genetic information [39], could be responsible for diversifying the repertoire of drugs that can be processed by CYP3A5. We have focused our analysis on five variants identified in sub-Saharan African populations. We note that this is not an exhaustive number, and there will likely be more identified because more African datasets will be sequenced in the future [10]. Understanding the relationship between African genetic diversity and drug response could provide valuable pharmacogenomic knowledge.

We observed that genetic diversity affects the binding of the enzyme to different drugs from the estimation of the binding free energy of ritonavir and artemether. In such regard, we believe that we have obtained reliable results based on previous enzyme assays where the $K_{i}$ value was estimated at 0.6–0.8 μM for ritonavir/CYP3A5 [40]. The equivalence in Gibbs free energy is −8.4 to −8.3 kcal mol^−1^ which is very close to what we have obtained for the wild type variant ( −8.72 kcal mol^−1^). Moreover, in addition to the genetic determinant, the binding capacity of CYP3A5 appears to be drug dependent. The strong polynomial correlation that we have noticed for ritonavir between the estimated free energy of binding and the pocket volume might be explained as follows. When the pocket volume is very low to optimally fit the ligand, atomic clashes may occur which renders the interaction less favorable. At specific ranges of pocket volume, shape complementarity is more favorable and ritonavir would fit properly; therefore, the steric clashes are less severe and the interaction energy is more favorable. When the volume increases too much, even with the presence of the drug, the Van der Waal’s interactions would not be satisfied which would penalize the interaction energy. Such a model could explain the behavior of bulky ligands where the stability depends on the flexible segments surrounding the catalytic pocket including the upper roof of the heme group and F/G loop. Artemether, however, does not seem to be impacted by the flexibility of these segments when interacting with the heme group because the drug is less bulky and the steric penalty would be less relevant.

The investigation of the major active tunnels for each variant has revealed the potential implication of the genetic variability in determining the access to the catalytic site. Although the access points to the tunnels on the surface of the protein can be broadly positioned at two sites, i.e., the upper roof segments nearby the F/G loop and the B/C loop, they, however, implicate different sets of amino acids at each gate provided with different chemical properties. Therefore, it affects the selectivity of the drug and the kinetic of association to the protein. Similar conclusions were obtained previously from molecular dynamics simulations of CYP2D6 variants [41].

R28C and Y53C variants have been characterized functionally using enzymatic assays following expression in *Escherichia coli* and purification protocols. They were found to decrease CYP3A5 catalytic activities for testosterone and Nifedipine respectively [25,42]. Nevertheless, we have not detected an obvious structural deviation during the 1.5
μs simulation but only significant alterations of the dynamics. More specifically for R28C, we have detected a significant reduction of catalytic pocket volume. This suggests that the functional implication of this variant might also be related to its capacity of inducing long-distance structural modulation of CYP3A5 unrelated to the membrane-binding function as suggested previously [42]. The R28 side chain is involved in stabilizing the local structure by establishing contacts with T27 and Q77 which would now be unreachable with the substitution to cysteine due to the short side chain.

The L82R variant has not been successfully expressed in *Escherichia coli* and the authors suggested that the lack of stability was the main reason for that [42]. However, we were not able to detect any major structural rearrangement but a significant variation of the major modes of dynamics. It might be possible that the folding process of CYP3A5 in the expression system misbehaved and such phenomena are often observed in *E. coli* [43]. L82R might seem to be a highly deleterious variant resulting from the burial of a charged residue in the core of the protein. However, a frequency of 2% has been detected in ethnically diverse samples from the USA that cause the same missense substitution L82R [42], which suggests a possible fixation in agreement with our findings.

Additionally, V238A, corresponding to variant rs542523237, appears to have the most significant effect on CYP3A5 because of its location on the F/G loop segment. The variant seems to affect the allostery of the protein, the pocket volume, and ritonavir binding.

Low protein levels of CYP3A5 were found to be related to *CYP3A5**3, *CYP3A5**6, and *CYP3A5**7 alleles [44] corresponding all to splicing defects/frameshift errors that result in the nonexpression of the gene product. *CYP3A5**3 (rs776746) was found to be the most relevant in non-African populations including individuals of European ancestry (0.82–0.95), Japanese (0.85), Chinese (0.65), Mexicans (0.75), and Pacific Islanders (0.65) [3] while only 4% of Africans are presented with *CYP3A5**3 [45]. In contrast, *CYP3A5**6 and *CYP3A5**7 prevalence is almost exclusive to populations of African ancestry with frequencies of 7–17% and 8% respectively. However, these were tested only in vitro in heterologous expression assays. Other missense-substitution SNPs of different prevalence in worldwide populations are likely to be associated with decreased expressions of CYP3A5 [42].

## 5. Conclusions

We have shown, in our work, that the genetic variability of *CYP3A5* results in a significant impact on the functional properties of the protein that include both plasticity-related and drug-binding related characteristics. Therefore, pharmacokinetic properties are significantly determined by the type of drug and the impact of the genetic diversity on the protein structure, resulting in a diversity of responses rather than manifesting solely in deleterious or non-deleterious outcomes. Our study highlights also the drug dependent phenotype relationship rather than the global effect related to the genotype. All the analyzed variants are rare, yet the heterogeneity of conformational properties revealed a significant effect. At least at the level of CYP3A5, we underlined the importance of the rare variants in determining the pharmacogenomics properties in populations from sub-Saharan Africa. It is also expected for other ADME proteins, particularly those in the cytochrome P450 family, to demonstrate the same properties. Consequently, it seems that the *’genetic diversity bottleneck for precision medicine’* is indeed a standing issue. It is still unclear, however, if the genetic variability will manifest into a significant clinical effect and the presented results for this paper should motivate a complete functional analysis of these variants. Variability in the amino acid sequence of CYP3A5 [42], other CYP3A subfamily members [46], and the orthologous CYP3A38 identified in Cattle (*Bos taurus*) [47] has shown a clear impact on the levels of activities of the enzyme. However, the clinical relevance of *CYP3A5* missense variants leading to amino acid substitution is difficult to determine due to numerous factors as shown from earlier studies. Missense variants of *CYP3A5* could make part of a larger haplotype with multiple polymorphic sites that could result in a combined effect on the phenotype. Moreover, heterozygous manifestation alongside another deleterious allele on the opposite chromosome can also obscure the effect of the missense variant. Additionally, pharmacokinetic studies associate increased or decreased levels of substrates/metabolites with the genotype. However, substrates of CYP3A5 overlap significantly [48] with CYP3A4 which can lead to conflicting outcomes [49]. In this regard, our study is highlighting the importance of several missense variants of *CYP3A5* and calls for a deeper analysis of their functional and clinical relevance using more reliable methods. The computational protocol can also be used to prioritize putatively functional variants for which phenotypic evidence is scarce. In the coming years, the expected drop in sequencing cost will lead to a better characterization of rare variants for ADME genes. As a result, it will be difficult to overlook the clinical implication of rare variants as key determinants of the pharmacogenomic landscape.

## Figures and Tables

**Figure 1 ijms-22-07786-f001:**
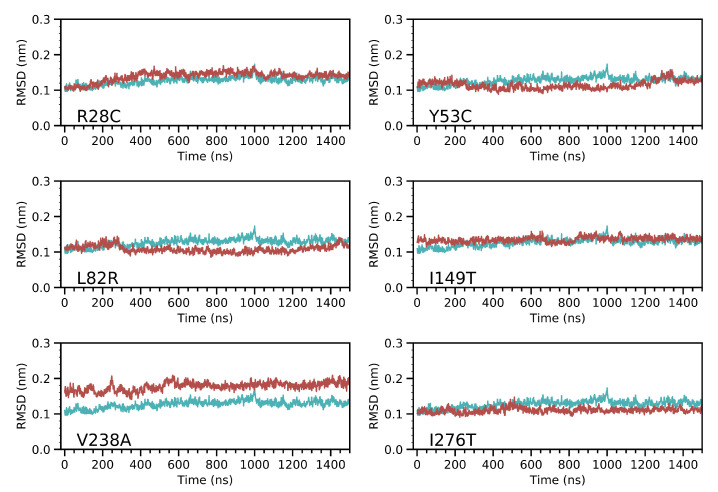
Analysis of the structural deviation CYP3A5 variants (**Red plots**) compared to the wild type form (**Cyan plot**). All the trajectory frames were first fitted to the crystal structure of the enzyme and the calculation was made using the coordinates of the backbone atoms. Loops 260–270 and 280–288 were omitted from the calculation.

**Figure 2 ijms-22-07786-f002:**
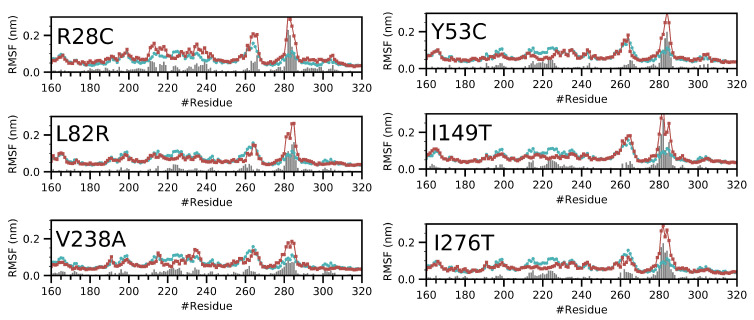
The fluctuation of the amino acid residues for CYP3A5 variants compared to the wild type form (**Blue plots**) and the variant forms (**red plots**). The bars indicate the difference of the fluctuation between the equivalent amino acids of the wild type and the variant. We only show the segment corresponding to amino acids 160–320. The RMSF profiles for the entire protein can be found in Appendix A.

**Figure 3 ijms-22-07786-f003:**
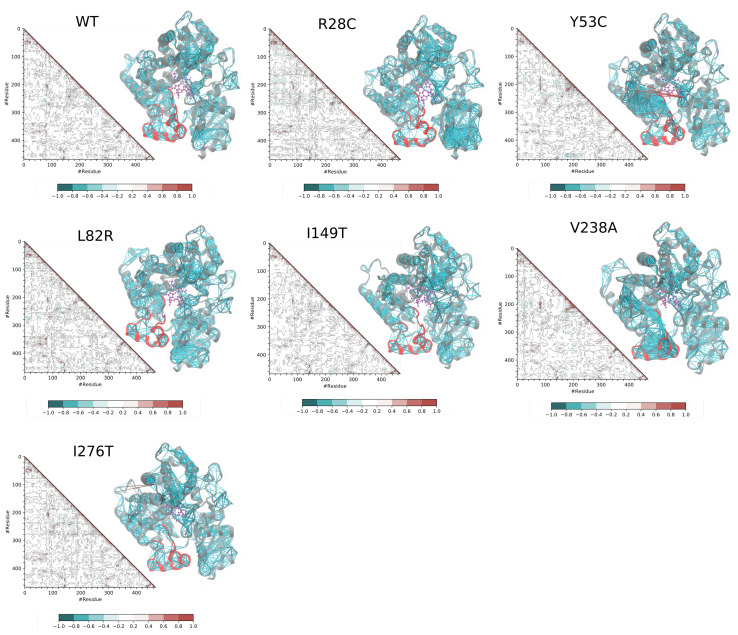
Dynamic Cross-Correlation Matrix (DCCM) analysis for the wild type and the variant forms. The data from each DCCM were mapped on the structure as a link between the pairs of residues established if DCC(i,j) is above 0.5 or below −0.5. The F/G loop is marked in red color on the structure. The index of the residues started at 0 and the reader should adjust with an offset of 25 to get the correct position in the sequence.

**Figure 4 ijms-22-07786-f004:**
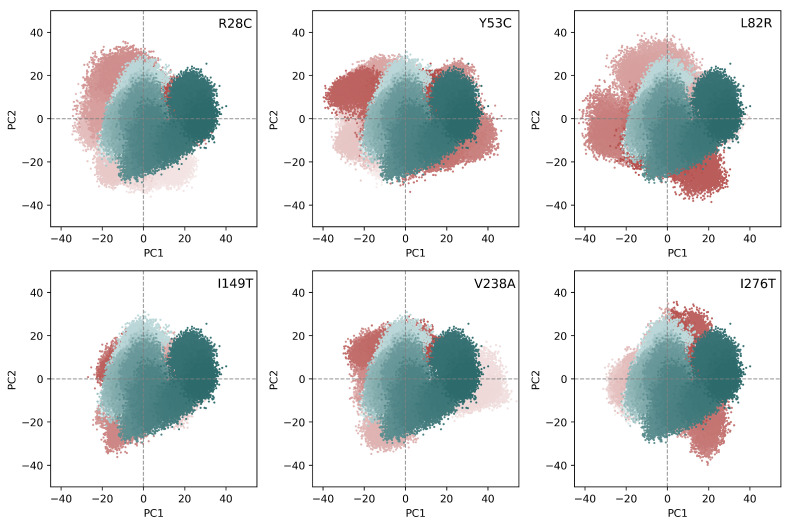
Projection of the $C_{\alpha }$ atoms coordinates on the first and second principal components. The blue and red colors indicate the snapshots of the wild type and the variant trajectories respectively. The calculation included only the snapshots collected after the 200 ns time.

**Figure 5 ijms-22-07786-f005:**
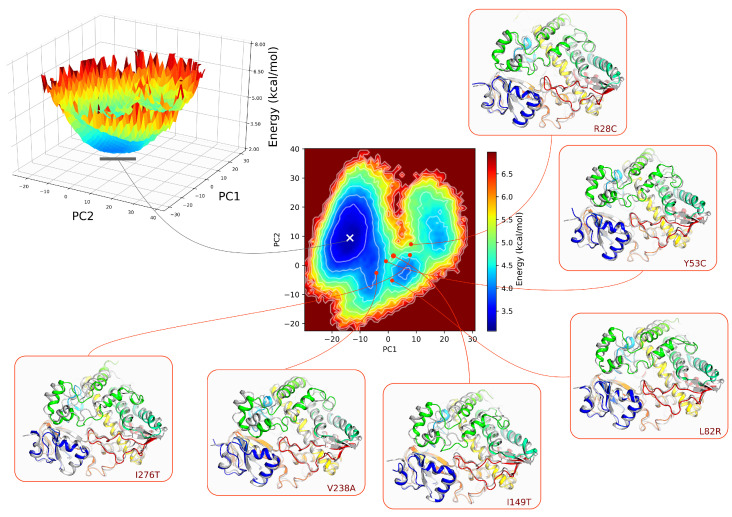
Free energy landscape of CYP3A5. PC1 and PC2 were calculated for all the $C_{\alpha }$ atoms excluding the highly flexible loops. The 3D representation of the FEL was projected in a 2D heatmap. Contour plots were also included to differentiate between the different levels of the energy. The minimum and the maximum energy values are 3.05 kcal mol^−1^ and 6.97 kcal mol^−1^. Each level in the contour plot corresponds to ±3.05 kcal mol^−1^. The position of the conformation at the global minimum of the wild type form is marked by the white cross. Structural fitting was performed between the conformations of the variants at their corresponding global minimum (**Rainbow color**) and the conformation of the wild type form (**Gray color**). The segments spanning the F helix, G helix, and F/G loops are colored in green and the B/C loop corresponds to the blue cartoon.

**Figure 6 ijms-22-07786-f006:**
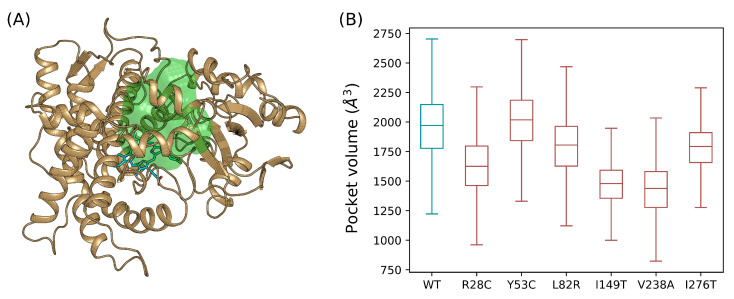
Calculation of the catalytic pocket volume of CYP3A5. (**A**) Location of the catalytic pocket calculated for the wild type form relative to the heme group. (**B**) The catalytic pocket volume was calculated for an ensemble of molecular dynamics snapshots at the equilibrium phase represented in boxplots.

**Figure 7 ijms-22-07786-f007:**
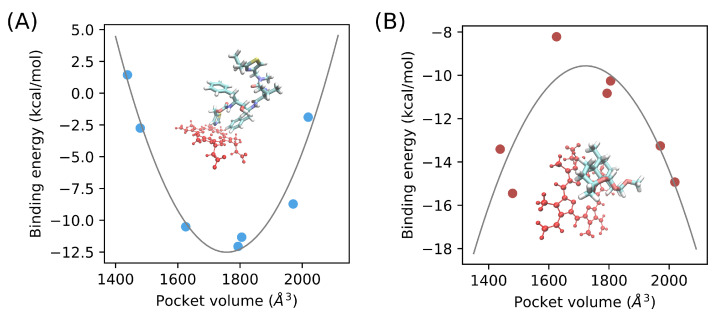
MM-GBSA energy as a function of the catalytic pocket volume for ritonavir (**A**) and artemether (**B**). The curves were fitted using a polynomial model of the form y=(x−a)(x−b)(x−c) where *a* = $1.33\times 10^{-4}$, *b* = −0.468, *c* = $3.98\times 10^{2}$ for ritonavir and *a* = $-6.25\times 10^{-5}$, *b* = 0.215, *c* = $-1.95\times 10^{2}$ for artemether. We also show the ligand spatial configuration relative to the heme group of CYP3A5.

**Figure 8 ijms-22-07786-f008:**
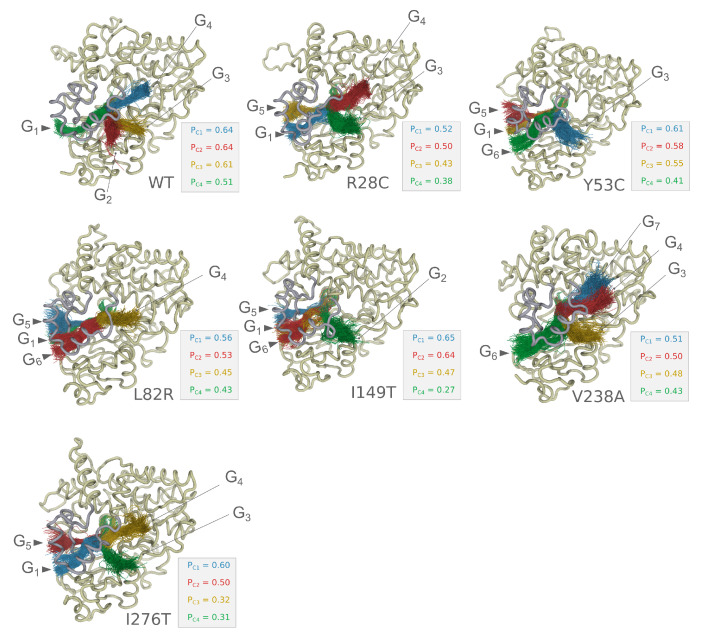
Analysis of tunnels for CYP3A5 variants. For each variant, we show tunnels with the best priority values encoded by the colors as follows: blue for the first rank, red for the second rank, yellow for the third rank, and green for the fourth rank. We also show the position of the gates for each variant.

**Table 1 ijms-22-07786-t001:** Frequency and amino acid substitution of missense variants characterized for *CYP3A5* gene from combined HAAD and KGP data. Nucleotide positions based on the *CYP3A5* NM_000777.5 transcript.

RS ID	Nucleotide	Amino Acid Variant	Frequency in HAAD + KGP
rs55817950	c.82C > T	R28C	0.0021
rs56244447	c.245T > G	L82R	0.0054
rs142823108	c.446T > C	I149T	0.0043
rs542523237	c.713T > C	V238A	0.001
rs145774441	c.827T > G	I276T	0.0021
rs72552791	c.158A > G	Y53C	0.000

**Table 2 ijms-22-07786-t002:** Binding free energy and corresponding energy terms calculated for CYP3A5/ritonavir and CYP3A5/artemether complexes.

Drug	Variant	$\Delta \;G_{\mathrm{binding}}$ (kcal/mol)	$\langle E_{\mathrm{pl}}^{\mathrm{int}}\rangle$ (kcal/mol)	−TΔS (kcal/mol)	$\Delta \;G_{\mathrm{solv}}$ (kcal/mol)
ritonavir	WT	−8.72	−58.46	11.01	38.72
	R28C	−10.51	−65.18	14.01	40.66
	Y53C	−1.89	−64.09	21.08	41.11
	L82R	−11.33	−60.87	12.69	36.84
	I149T	−2.75	−57.84	15.29	39.79
	V238A	+1.44	−55.97	19.45	37.96
	I276T	−12.06	−60.85	13.58	35.20
artemether	WT	−13.26	−29.57	7.75	8.55
	R28C	−8.22	−33.16	12.08	12.85
	Y53C	−14.93	−29.31	6.13	8.24
	L82R	−10.26	−30.71	10.36	10.08
	I149T	−15.45	−31.0	7.66	7.88
	V238A	−13.41	−29.59	8.43	7.74
	I276T	−10.83	−29.80	9.47	9.49

## Data Availability

All the raw data and the code used to make the analysis and figures of this paper are available online from the Zenodo repository under the DOI 10.5281/zenodo.4548257.

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
