# Peer review of "Single Nucleotide Polymorphism Induces Divergent Dynamic Patterns in CYP3A5: A Microsecond Scale Biomolecular Simulation of Variants Identified in Sub-Saharan African Populations"

_ijms, 2021, doi:10.3390/ijms22157786_

Round 1
Reviewer 1 Report
The article by Othman and colleagues is a nice representation of the emerging field of precision medicine, and especially the pharmacogenomics of CYP3A5 on which consistent data are not widely available. Only few comments: - Please double check the abstract for all sentences to be in past tense. - Please provide a better explanation of docking concept in section 2.4. - Please include some of the figures in supplementary files in the text, particulary figures 1 and 5. - Please perform a language and grammar check.
Author Response
The modifications made at the request of the reviewer are marked in the manuscript and indexed in the footnote.
Hereby, we provide a point-by-point response to the reviewer’s comments.
1 - Please double check the abstract for all sentences to be in past tense.
We have modified all the sentences to be in the past tense.
2 - Please provide a better explanation of the docking concept in section 2.4.
We have added a new paragraph to the beginning of section 2.4 that explains the principle of the docking.
“ The molecular docking approach is used to predict the bound form between a ligand and a target (commonly known as a receptor) when a reliable experimental structure of the native complex is not available. Docking algorithms sample the translational and rotational degrees of freedom of a ligand over a 3D domain-space of the receptor based on an energy scoring function.“
3 - Please include some of the figures in supplementary files in the text, particulary figures 1 and 5.
We agree with the reviewer that the charge of the figure needs to be alleviated. We have included figures 1, 5, and 7 in supplementary data.
4 - Please perform a language and grammar check.
The language and grammar were checked again.
Reviewer 2 Report
Title:
Single nucleotide polymorphism induces divergent dynamic patterns in CYP3A5: a microsecond scale biomolecular simulation of variants identified in Sub-Saharan African populations
In this original article, the authors have investigated the genetic data of CYP3A5 from 458 individuals from sub-Saharan Africa to conduct a structural bioinformatics analysis. Five missense variants were modeled and microsecond scale molecular dynamics simulations were conducted for each, as well as for the CYP3A5 wildtype, and the Y53C variant, which has a known deleterious impact on enzyme activity and concluded that rare variants genetic diversity in affecting the gene product functionally and could have a significant impact on precision medicine applications. Please address the following questions or suggestions.
Major Comments:
- The authors can elucidate CYP3A5 expression level and the role of SNPs in Sub-Saharan African populations compared to the other ethnic groups in the discussion section.
- The extensive role of CYP3A4 in drug metabolism reflects the plasticity of the substrate free enzyme to enlarge its active site and accommodate very large substrates has been reported. The authors can highlight, whether the structure of CYP3A5 ritonavir complex differs substantially from that of the CYP3A4 ritonavir.
- Why the authors used only the 1000 Genomes reference panel? Is there is any study on the GWAS that attempts to identify commonly occurring genetic variants in artemether and ritonavir in Sub-Saharan African populations.
- Certain studies have reported that individual CYP3A5 polymorphisms were not significantly associated with differences in pharmacokinetics of the substrates. (https://www.mdpi.com/2227-9059/8/4/94). Add a note/table on reviewing studies focusing on CYP3A5 genotype-pharmacokinetic correlation studies.
- The study has mentioned that 280 drugs are substrates for CYP3A5. What are the major drug substrates for CYP3A5 other than artemether and ritonavir? Add notes regarding them into article including clinically relevant inducers and inhibitors of CYP3A5 since the study has reported that impact of drug binding is drug dependent.
- The study reports that genetic variability of CYP3A5 resulted in a significantly impacting the functional properties of the protein, however the clinical impact of the genotype could not be perceived. Please add notes in discussion/conclusion regarding how the results of the current study could be scaled up to assess the potential impact of CYP3A5 genotype variants on pharmacokinetics of drugs such as ritonavir.
- In table 1, please add allelic information also regarding each variant.
- CYP3A5 is a highly polymorphic gene with ~29 total variants. But only 5 missense variants were specifically modelled with further MDS studies in the current study. Why were only these 5 variants considered? Were they more frequent or whether any studies have shown that there is computation data/clinical relevance? Please give details regarding the relevance of selecting these CYP3A5 variants.
Minor Comments:
- In the Abbreviations Section VEP, KGP – Correct it
- Reference no 7, 8,10, 24 and 37 – Check the volume and page no
I can recommend this manuscript for publication in IJMS Journal after above-mentioned revisions.
Author Response
The modifications made at the request of the reviewer are marked in the manuscript and indexed in the footnote.
Hereby, we provide a point-by-point response to the reviewer’s comments.
1- The authors can elucidate CYP3A5 expression level and the role of SNPs in Sub-Saharan African populations compared to the other ethnic groups in the discussion section.
We have added a new paragraph to the end of the discussion to cover the requested modifications.1
“Low protein levels of CYP3A5 were found to be related to CYP3A5*3, CYP3A5*6 and CYP3A5*7 alleles (pmid11279519) corresponding all to splicing defects/frameshift errors that result in the non-expression of the gene product. CYP3A5*3 (rs776746) was found to be the most relevant in non-African populations including individuals of European ancestry (0.82–0.95), Japanese (0.85), Chinese (0.65), Mexicans (0.75), and Pacific Islanders (0.65) (pmid22407409) while only 4\% of Africans are presented with CYP3A5*3 (MASIMIREMBWA2014971). In contrast, CYP3A5*6 and CYP3A5*7 prevalence is almost exclusive to populations of African ancestry with frequencies of 7–17% and 8% respectively. However, these were tested only in vitro in heterologous expression assays. Other missense-substitution SNPs of different prevalence in worldwide populations, are likely to be associated with decreased expressions of CYY3A5.“
2- The extensive role of CYP3A4 in drug metabolism reflects the plasticity of the substrate free enzyme to enlarge its active site and accommodate very large substrates has been reported. The authors can highlight whether the structure of CYP3A5 ritonavir complex differs substantially from that of the CYP3A4 ritonavir.
We have made a comparison with the catalytic pocket volume of CYP3A4. Essentially, our data showed increased volume compared to CYP3A4. We have added a new paragraph to unwrap all this in section 3.6.
“The CYP3A subfamily was shown to be promiscuously allowing the accommodation of a large repertoire of drugs in the catalytic pocket of the enzymes. The wild-type pocket volume range of CYP3A5 calculated in our study confirmed such property. Our results also show higher volume pocket values reported previously for CYP3A4 evaluated to 1386 Angstroms and 520 Angstroms from two different studies. These values, however, were reported based on single-point calculations for the unbound structure of CYP3A4. Another calculation obtained a value of volume-pocket of 765 Angstroms of CYP3A4 bound to ritonavir which is almost two times less than the lowest volume calculated to any of CYP3A5 variants in complex with the same drug.“
3- Why the authors used only the 1000 Genomes reference panel? Is there is any study on the GWAS that attempts to identify commonly occurring genetic variants in artemether and ritonavir in Sub-Saharan African populations.
Thank you for the questions. We used the 1000 Genomes data as it was the largest collection of available African sequences (from 507 African individuals). These data were jointly called with high coverage data from the H3Africa/GSK ADME consortium (458 African individuals) to enable optimal discovery of variants (da Rocha et al - https://doi.org/10.3389/fphar.2021.634016). We have added to methods: “Variants were identified in jointly called HAAD and 1000 Genomes datasets.”
On the assessment of ritonavir and artemether in sub-Saharan Africans: we have added to the introduction:
“The CYP3A5*3 splice variant has been studied in African populations regarding ritonavir (PMID 27142945) and artemether (PMID 28934955) drug response, but studies of CYP3A5 missense variants in these populations remain scarce.”
4- Certain studies have reported that individual CYP3A5 polymorphisms were not significantly associated with differences in pharmacokinetics of the substrates. (https://www.mdpi.com/2227-9059/8/4/94). Add a note/table on reviewing studies focusing on CYP3A5 genotype-pharmacokinetic correlation studies.
We think it is pertinent to describe the state of the art of conflicting results in pharmacogenetics/pharmacogenomics studies about CYP3A5. We think that the systematic review and meta-analysis by Khan et al, (2020), is giving the same extensive details asked by the reviewer. We have added a note about this in the text and referred to the paper in the introduction section. Moreover, the Article by Saiz-Rodríguez was cited in reply to comment 6.
“There have been some conflicting studies about the genotype-phenotype association of CYP3A5. A recent systematic review and meta-analysis showed that CYP3A5 genotype does not correlate uniformly with pharmacokinetics properties of tacrolimus (pmid31902947). Variant levels of significance response have been noted between different populations as well as within populations of the same ethnic composition.”
5- The study has mentioned that 280 drugs are substrates for CYP3A5. What are the major drug substrates for CYP3A5 other than artemether and ritonavir? Add notes regarding them into article including clinically relevant inducers and inhibitors of CYP3A5 since the study has reported that impact of drug binding is drug dependent.
We have added a list of 287 drugs of CYP3A5 as well as their types (substrate, inhibitor, inducer) in supplementary material 1.
6- The study reports that genetic variability of CYP3A5 resulted in significantly impacting the functional properties of the protein, however the clinical impact of the genotype could not be perceived. Please add notes in discussion/conclusion regarding how the results of the current study could be scaled up to assess the potential impact of CYP3A5 genotype variants on pharmacokinetics of drugs such as ritonavir.
We have pointed to the fact that many functional studies, conducted for CYP3A5 and other members of the same family have shown the relevance of amino acid sequence in determining the enzymatic activity. We have put a note in the conclusion section to discuss the point thoughtfully.
“Variability in the amino acid sequence of CYP3A5, other CYP3A subfamily members as well as the orthologous CYP3A38 identified in Cattle (Bos taurus) has shown a clear impact on the levels of activities of the enzyme. However, the clinical relevance of CYP3A5 missense variants leading to amino acid substitution is difficult to determine due to numerous factors as shown from earlier studies. Missense variants of CYP3A5 could make part of a larger haplotype with multiple polymorphic sites that could result in a combined effect on the phenotype. Moreover, heterozygous manifestation alongside another deleterious allele on the opposite chromosome can also obscure the effect of the missense variant. Additionally, pharmacokinetic studies associate increased or decreased levels of substrates/metabolites with the genotype. However, substrates of CYP3A5 overlap significantly with CYP3A4 which can lead to conflicting outcomes. In this regard, our study is highlighting the importance of several missense variants of CYP3A5 and calls for a deeper analysis of their functional and clinical relevance using more reliable methods. The computational protocol can also be used to prioritize putatively functional variants for which phenotypic evidence is scarce.“
7- In table 1, please add allelic information also regarding each variant.
Allelic information was added to the table according to the HGVS nomenclature for each of the presented variants.
8- CYP3A5 is a highly polymorphic gene with ~29 total variants. But only 5 missense variants were specifically modelled with further MDS studies in the current study. Why were only these 5 variants considered? Were they more frequent or whether any studies have shown that there is computation data/clinical relevance? Please give details regarding the relevance of selecting these CYP3A5 variants.
Our study is a secondary analysis of the data described originally by da Rocha et al (https://doi.org/10.3389/fphar.2021.634016). The study reports on the largest cohort from a sub-Saharan African population using the whole genome sequencing method. We have processed all the missense variants identified by this study for CYP3A5. We are certain that the genetic makeup of missense variants for CYP3A5 is not limited to these 5 variants, a point that we have emphasized in the conclusion. As more genomic data from Africa will be available in the future, the repertoire of CYP3A5 genetic variability will be enriched. However, even including all the 5 rare variants identified in our dataset, is in line with the objective of the study which is the evaluation of the structural impact and the extent of the genetic diversity on the protein conformational properties.
We have added to discussion:
“We have focused our analysis on five variants identified in sub-Saharan African populations. We note that this is not an exhaustive number, and there will likely be more identified as more African datasets are sequenced in future (https://www.frontiersin.org/article/10.3389/fphar.2021.634016). Understanding the relationship between African genetic diversity and drug response could provide valuable pharmacogenomic knowledge.”
Minor Comments:
1. In the Abbreviations Section VEP, KGP – Correct it
This is now corrected in the submitted version.
2. Reference no 7, 8,10, 24 and 37 – Check the volume and page no
We have verified the information about the volume and the page numbers in each one of these articles. We found that all the above references are complying with the recommendation of the publisher about how to cite the attributes of each article. Citing a one-page number might also be recommended when the paper is only available online.
Round 2
Reviewer 2 Report
No comments.